# GN-Transformer: Fusing AST and Source Code information in Graph Networks

## Abstract

As opposed to natural languages, source code understanding is influenced by grammar relations between tokens regardless of their identifier name. Considering graph representation of source code such as Abstract Syntax Tree (AST) and Control Flow Graph (CFG), can capture a token's grammatical relationships that are not obvious from the source code. Most existing methods are late fusion and underperform when supplementing the source code text with a graph representation. We propose a novel method called GN-Transformer to fuse representations learned from graph and text modalities under the Graph Networks (GN) framework with attention mechanism. Our method learns the embedding on a constructed graph called Syntax-Code Graph (SCG). We perform experiments on the structure of SCG, an ablation study on the model design and the hyperparamaters to conclude that the performance advantage is from the fusion method and not the specific details of the model. The proposed method achieved state of the art performance in two code summarization datasets and across three metrics.

## 1 Introduction

Code summarization is the task of generating a readable summary that describes the functionality of a snippet. Such task requires a high-level comprehension of a source code snippet thus it is an effective task to evaluate whether a Deep Learning Model is able to capture complex relations and structures inside code. Programming languages are context-free formal language, an unambiguous representation, Abstract Syntax Tree (AST), could be derived from a source code snippet. A parse tree based representation of code is precise and without noise. An AST accurately describes the structure of a snippet and relationships between tokens which provides valuable supplementary information for code understanding.

Using graph representations of source code has been the focus of multiple methods that perform code summarization. For example, Alon et al. (2019a) encoded AST paths between tokens and aggregated them by an attention mechanism. Huo et al. (2020) used CNN and DeepWalk on Control Flow Graph (CFG) to learn statements representations. LeClair et al. (2020) applied Graph Convolutional Networks (GCNs) to learn AST representation. These methods proposed different ways to

Figure 1: Examples of generated summarizations on Java (red blocks) and Python (blue blocks) test set. Transformer is a vanilla Transformer, Transformer (full) implements relative positional encoding and copy mechanism from Ahmad et al. (2020).

extract AST features, however the *cross-modal interaction* (Veličković, 2019) is very limited since the AST and code features are independently extracted by separate models then simply concatenated or summed.

In this paper we propose a novel architecture **GN-Transformer** shown in Figure 2 to fuse Graph information with an equivalent sequence representation. In summary:

- We extend Graph Networks (GN) (Battaglia et al., 2018) to a novel *GN-Transformer* architecture that is a sequence of GN encoder blocks followed by a vanilla Transformer decoder.
- We propose a novel method for early fusion of the AST representation and that of a code snippet sequence called *Syntax-Code Graph* (SCG)
- We evaluate our approach on the task of code summarization and outperform the previous state of the art in two datasets and across three metrics.

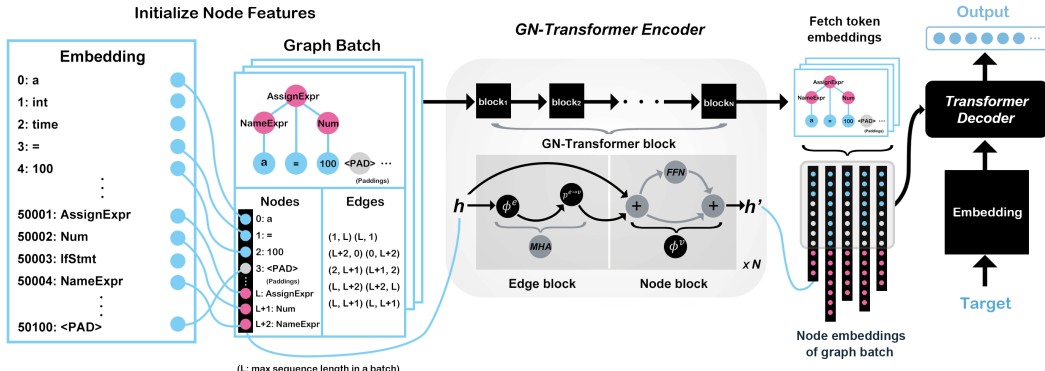

Figure 2: Overall structure of our model. Encoder consists of multiple GN-Transformer blocks. We denote '+' as a residual connection followed by a normalization layer. In 'Node embeddings of graph batch', each black bar represents the nodes embedding of a graph in the input batch. Blue dots represent token nodes, grey dots denote padding. Nodes embedding in the grey box are fetched as input to the decoder and AST nodes embedding (red dots) are discarded.

We evaluated our model on Java and Python datasets used by Ahmad et al. (2020). We compared our results to those of Ahmad et al. (2020). Two qualitative results are presented in Figure 1. We make available our code, trained models and pre-processed datasets in our supplementary package, and we will open-source it after the review process concludes.

## 2 FUSING GRAPH AND SEQUENCE INFORMATION

Previous methods consider sequences and graphs as two modalities that are processed independently. For a sequence, recurrent architectures such as RNNs, LSTMs, GRUs are commonly used. CNNs have also been applied on sliding windows of sequences (Kim, 2014). Transformers (Vaswani et al., 2017) became a popular choice for sequences in recent years. For graph data, spectrum-based methods like GCNs (Bruna et al., 2014) capture graph structure through a spectrum. Non-spectrum methods like GraphSAGE (Hamilton et al., 2017) aggregates information from neighboring nodes using different aggregators, GAT (Veličković et al., 2018) introduced attention mechanism to aggregate neighboring information. Early fusion of multiple modalities is a challenging task. As a result, late fusion methods are used when considering multi-modal information in code summarization tasks. The cross-modal interactions are less efficient in late fusion as compared to early fusion Veličković (2019). In Section 2.1 we discuss early fusion approaches of code sequence with an AST. In Section 2.2 we discuss representing sequence in a graph.

### 2.1 EARLY FUSION OF SEQUENCE AND GRAPH

For early fusion of a sequence and a graph, it is common to represent them under a single unified representation and input to a deep learning model. Random walks (Perozzi et al., 2014) and

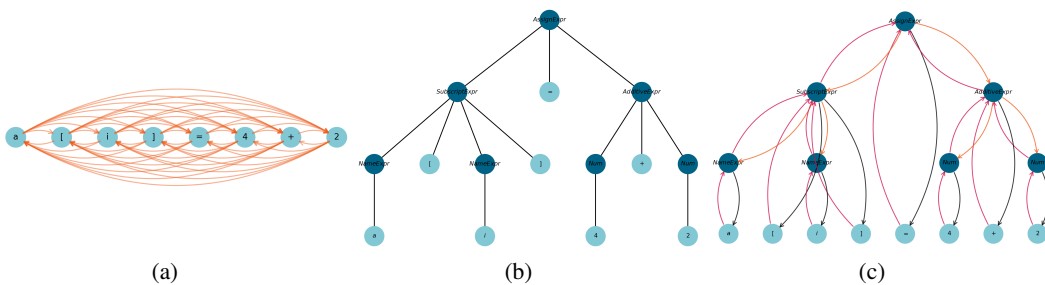

|       |       |       |
|:-----:|:-----:|:-----:|
|  (a)  |  (b)  |  (c)  |

Figure 3: (a) Simplest fully-connected graph representation of sequence. Each node corresponds to a token. Self-loops are omitted. (b) Deep blue nodes are AST for the statement 'a[i]=4+2', shallow blue nodes shows the correspondence between AST and source code. (c) Standard SCG structure. It preserves the AST structure with additional edges between AST nodes and tokens.

structure-based traversal (SBT) (Hu et al., 2018a) demonstrated the advantages of flattening a graph structure into a sequence. SBT converts AST into a sequence that can be used by seq2seq architectures. However, structural information contained in the AST is lost when flattened into an unstructured sequence.

The main motivation of previous methods to flatten a graph structure into a sequence instead of the opposite is due to the power of sequence models when compared to their graph counter-parts. Recent advances in general frameworks such as Graph Networks (Battaglia et al., 2018), Message-Passing Neural Networks (MPNN) (Gilmer et al., 2017) proposed a unified graphical representation of data. Battaglia et al. (2018) proposed an extension and generalization of previous approaches that can learn a graphical representation of data under a configurable computation unit, GN block. We discuss our GN block based encoder in Section 4.2.

There are several benefits in graph representation. Firstly it can contain different information sources with arbitrary relational inductive bias (Battaglia et al., 2018). Secondly, a graph representation can have explicit relational inductive biases among graph entities. Thirdly, flexible augmentation of graph structure of input through expert knowledge. Additionally, better combinatorial generalization (Battaglia et al., 2018) due to the reuse of multiple information sources simultaneously in a unified representation. Finally, measures for analyzing performance through graph structures could be naturally incorporated, such as average path length $L$ and clustering coefficient $C$. We discuss it in better detail in Section 3.2.

## 2.2 GRAPH REPRESENTATIONS FOR SEQUENCE

In Graph Networks, defining the graph structure of input data is the main way to introduce relational inductive biases (Battaglia et al., 2018). Relational inductive biases impose constraints on the relationship among entities. In Graph Networks, the assumption that two entities have a relation or interaction is expressed by an edge between the corresponding entity nodes. The absence of an edge expresses the assumption of isolation which means no relation or direct influence of two nodes.

Sequences are unstructured data in which relationships are implicit. Implicit relationships can be represented with each token as a graph node that is fully connected with all other nodes. Such representation allows each node to interact with every other node with no explicit assumptions on isolation. Thus it allows the model to infer the relationships between them implicitly. Transformers could be regarded as inferring on such a fully connected graph, see Figure 3(a). Each token in an input sequence corresponds to a node. The relationship between tokens is thus represented by attention weights, high attention values correspond to strong interaction while low attention means isolation.

It is less efficient for a model to learn to infer relationships without any explicit assumptions of interactions and isolation. AST provides precise information about interaction and isolation among tokens in source code since it's a representation without noise on how the tokens interact during the execution of a code snippet. Thus a natural way of utilizing information from an AST is fusing

the graph structure with an input sequence. We can find an explicit mapping between tokens in a sequence and nodes in the AST through the scope information provided by a parser for a given programming language. We are then able to find relations thus build edges between a sequence and a graph. We'll further discuss the graph structure of input data and how we fuse AST with it in Section 3. The idea of fusing a sequence and a graph can be extended to broader cases apart from AST and code snippet when the mapping is not explicit. There are techniques that are applied in knowledge graphs to find a mapping between entities in knowledge graphs and words in sequence (Peters et al., 2019). For our problem statement, we make the reasonable assumption that an explicit mapping between the sequence and a graph can be provided.

## 3 GRAPH STRUCTURE OF AST AND CODE

First, we propose a simple joint graph representation of source code and AST called standard Syntax-Code Graph (SCG) in Section 3.1. In Section 3.2 we discuss the influence of the structure of SCG and the existence of a theoretically optimal graph structure.

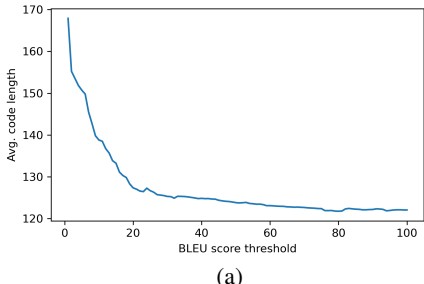
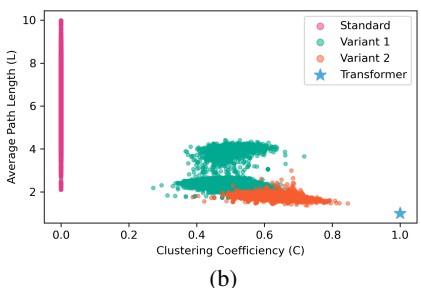

(a)            (b)

Figure 4: (a) Average code length of subsets of samples under different sentence BLEU score thresholds on the test set of Java dataset. (b) The average path length and clustering coefficient of our SCG structures are calculated on the test set of Java dataset and Transformer.

### 3.1 SYNTAX-CODE GRAPH

Standard SCG consists of AST nodes directly from AST and token nodes created for each code snippet token or simply referred to the remainder of the paper as *token node*. The attribute of an AST node is the type on AST such as "NameExpr". The attribute of a token node is the identifier name such as "a", "int", "+". Standard SCG preserves the AST structure and introduces additional edges that connect token nodes with their direct parent node in AST. The direct parent node for a token node is found through the scope information for AST nodes. Taking the statement 'a[i]=4+2' as an example, the AST is shown in Figure 3(b). The scope of an AST node is determined by the positional mark given by a compiler. The positional mark of the AST node 'AdditiveExpr' is *line 1, col 6 ∼ line 1, col 8* which corresponds to '4+2', so the scope covers token nodes '4', '+', and '2'. However, a token may be covered by the scopes of multiple AST nodes, we only connect the token with its direct parent in AST. The direct parent of a token is the deepest AST node among all AST nodes which scope covered the token. Figure 3(c) shows standard SCG.

Standard SCG directly builds on top of the AST structure without introducing any additional relational assumptions. Thus it objectively reflects the program structure depicted by a compiler.

### 3.2 OPTIMAL SYNTAX-CODE GRAPH STRUCTURE

The graph structure of SCG plays a critical role in our method. You et al. (2020) represented neural network structure as relational graphs. Message passing in a relational graph is analogous to information propagation in Graph Networks. SCG can be directly defined as the relational graph of our model. Node features could be defined as node embedding in SCG. Likewise, the message function is node-wise FFN and the aggregation function is MHA. As a result, the graph structure of SCG

will influence the performance of the model. The standard SCG structure discussed in Section 3.1 is efficient but may not be optimal.

Qualitatively, we observed two problems of standard SCG. The first problem is the existence of long range dependencies. It is hard to pass information between nodes separated by long paths in SCG. In Graph Networks, each node is executing an information propagation and aggregation simultaneously with neighboring nodes. For example, One graph neural networks (GNN) layer could be regarded as executing one turn of information propagation and aggregation. In Figure 3(c), it would require three GNN layers for information to propagate from leaf node 'a' to root node 'AssignExpr'. Thus it is difficult to propagate information between nodes in a large AST. The second problem is the isolation among token nodes. Source code token nodes can only indirectly interact with each other through AST nodes. Taking '4+2' in Figure 3(c) as an example, there is no direct edge between token nodes '4' and '2', they can only indirectly interact through the AST node 'AdditiveExpr'. However, the information that will be passed from 'AdditiveExpr' to '4' and '2' is identical. Therefore, it makes it difficult for the token nodes to learn more complex relationships with other token nodes when they are under the same expression. We analyzed the test set of the Java dataset and present the results in Figure 4(a). Longer code usually contains more long range dependencies and complex expressions that correspond to the above two problems.

Hand-engineering graph structure with expert knowledge may alleviate these problems thus improve performance. However, introducing redundant edges could be lead to performance degradation. We perform two experiments on opposite sides of the spectrum in our graph structure to examine the problems above. **Variant 1** added shortcut edges between a code token and AST nodes with the scope covered by that token thus the distance between all AST nodes to token nodes within its scope was shortened to 1. **Variant 2** makes token nodes fully connected, thus there is no isolation at all. We reference to those variants as *Variant 1* and *Variant 2* in the remainder of this text. More details on the two variants are described in Appendix D. Both variants failed to improve performance due to loss of structural information and redundant connections respectively, leading us to conclude that a trade-off of both is required. We discuss it in further detail at Section 5.2.

The optimal graph structure was also analyzed through quantitative experiments. You et al. (2020) proposed to measure the effectiveness of the message passing in a relational graph by the Average path length ($L$) and the clustering coefficient ($C$). The authors claim that the optimal structure is a balance between the value of $C$ and $L$. Figure 4(b) shows $L$ and $C$ for our variants, a *Standard* AST, and that of a fully-connected relational graph such as in a Transformer which could be formulated into a relational graph by the same way of ours with different relational graph structure. A vanilla Transformer encoder has a maximum $C$ of 1 and a minimum $L$ of 1. Our standard SCG, at the other extreme, has a minimum $C$ of 0 and a relatively high $L$ due to its tree structure. The average $L$ of standard SCG in Java and Python dataset is 6.28 and 6.35 respectively. Figure 4(b) shows that we only explored a small area of the entire design space of the possible graph structure for various $L$ and $C$. You et al. (2020) proposed that the optimal structure is usually located around a "sweet spot" in design space between extreme cases of the tree and fully-connected structures. Model performance will improve nearing this "sweet spot". Thus we conclude that there is a large potential of improving model performance by improving the input structure either through hand-engineering or additional rules tailored for the specific problem domain.

## 4 MODEL ARCHITECTURE

Our model follows the generic Transformer encoder-decoder architecture. The encoder is an extended architecture of Graph Networks with multiple GN blocks. The decoder is a vanilla Transformer decoder. The overview of our architecture is presented in Figure 2. We'll introduce our overall structure in Section 4.1. Then propose our Graph Networks based encoder in Section 4.2.

### 4.1 ENCODER-DECODER ARCHITECTURE

The encoder consists of a stack of $N$ GN-Transformer blocks which are derived from GN blocks, the main computation unit in Graph Networks, it takes a graph as input and returns a graph as output (Battaglia et al., 2018). Each block in our model implements a multi-head attention (MHA) sublayer and a feed-forward network (FFN) sublayer equivalent to a Transformer encoder layer.

Encoder accepts a graph $\mathcal{G}$ with the node features $h$ as input. $\mathcal{G} = \{V, E\}$ where $V$ is node set and $E$ is the edge set. $h \in \mathbb{R}^{|V| \times d_{model}}$ are the node features in $\mathcal{G}$, where $d_{model}$ is the input and output dimension of the encoder. The node features in the input graph are initialized through an embedding layer. AST and token nodes fetch an embedding vector according to their type and identifier names respectively. For our implementation, we handle token code identifiers and AST node type separately when performing an embedding lookup, as to avoid naming conflicts between the two. Encoder outputs the graph with the updated node features $h'$. Feature vectors of only token nodes are fetched as the decoder input. Feature vectors of AST nodes are discarded and the token embeddings are padded for batching (see Figure 2). Residual connections and layer normalization are used in Transformers and are also employed within each block in our model. Moreover, due to equivalence in network architectures, parameters number of our model is the same as the vanilla Transformer with the same configurations.

### 4.2 GN-TRANSFORMER BLOCKS

We extend GN blocks as proposed by Battaglia et al. (2018) and define MHA and FFN in this context and we call it a GN-Transformer block. One GN-Transformer block will execute one round of information propagation and aggregation with the neighboring nodes that updates the node and edge attributes on a graph. This is done by two sub-blocks an edge block that updates edge attributes, then a node block that updates node attributes. Notice that for our problem the initial edge attribute matrix of a SCG is the presence of a connection between two nodes.

**Edge block** $E$ is updated by an **edge update function** $\phi^e(h_i, h_j) = \frac{h_i W_\gamma^Q (h_j W_\gamma^K)^T}{\sqrt{d_k}}$ where $h_i$ and $h_j$ are attributes of node $i$ and $j$ respectively and $E_{ij}^{\prime(\gamma)}$ is the updated attribute of edge from node $i$ to $j$ through attention head $\gamma$. Each GN-Transformer block has $H$ attention heads. $W_\gamma^K \in \mathbb{R}^{d_{model} \times d_k}$ and $W_\gamma^Q \in \mathbb{R}^{d_{model} \times d_k}$ are parameter matrices for attention head $\gamma$.

And an **edge aggregate function** $p^{e \to v}(E_i') = Concat(head_i^{(1)}, ..., head_i^{(H)})W^O$ with $head_i^{(\gamma)} = \sum_{j \in N_i} h_j W_\gamma^V \alpha_{ij}^{(\gamma)}$ which aggregates updates of edges with $\alpha_{ij}^{(\gamma)} = \sigma(E_i^{\prime(\gamma)})_j$, the attention from node $i$ to the set of neighboring nodes $N_i$, $\sigma(\cdot)$ is softmax function, $W_\gamma^V \in \mathbb{R}^{d_{model} \times d_v}, W^O \in \mathbb{R}^{H d_v \times d_{model}}$ are parameter matrices.

A **Node block** $V$ updated by a **node update function** $\phi^v(h_i, p_i^{e \to v}) = FFN(\bar{h}_i) + \bar{h}_i$ where $\bar{h}_i = h_i + p_i^{e \to v}$ and $FFN(x) = max(0, xW_1 + b_1)W_2 + b_2$.

$\phi^e$ together with $p^{e \to v}$ define the MHA. $\phi^v$ implemented a node-wise FFN with residual connections similar to point-wise FFN in a Transformer. For our experiments, we also use dropout and layer normalization the same as Vaswani et al. (2017).

## 5 EXPERIMENT

We evaluated our model in two code summarization datasets. Additionally, we perform experiments on the hyperparameters, model structure, and variants of the graph structure. Our experiment settings are presented in Section 5.1. Results are presented and analyzed in Section 5.2.

### 5.1 SETTINGS

The experiments are conducted on a Java dataset (Hu et al., 2018b) and a Python dataset (Barone & Sennrich, 2017). We used JavaParser[1] to extract the AST and javalang[2] for parsing Java source code, python ast[3] to parse and get the AST for Python.

We chose a Transformer as our main comparison baseline which achieved state of the art in the two datasets by Ahmad et al. (2020). Ahmad et al. (2020) proposed a base model which is a vanilla Transformer and a full model that implements relative position embedding and copy mechanism.

---

[1]https://javaparser.org/
[2]https://github.com/c2nes/javalang
[3]https://docs.python.org/3/library/ast.html

Table 1: Overall results on Java and Python datasets.

| Models | Java | | | Python | | |
|---|---|---|---|---|---|---|
| | BLEU | METEOR | ROUGE-L | BLEU | METEOR | ROUGE-L |
| DeepCom [4] | 39.75 | 23.06 | 52.67 | 20.78 | 9.98 | 37.35 |
| Dual Model [5] | 42.39 | 25.77 | 53.61 | 21.80 | 11.14 | 39.45 |
| Transformer [6] | 43.99 | 26.40 | 53.30 | 31.22 | 18.56 | 44.22 |
| Transformer (full) [6] | 44.84 | 26.89 | 54.80 | 32.79 | 19.63 | 46.51 |
| GN-Transformer (Ours) | **45.48** | **26.91** | **55.29** | **33.31** | **19.66** | **46.56** |

We reproduced their results on our preprocessed datasets. Our preprocessed datasets are composed of the source code, corresponding AST, and a text summary. We also compared our method with the results of other baselines reported in Ahmad et al. (2020). Additional details about the datasets and preprocessing are in Appendix B. We used the same metrics that are reported by all baselines to evaluate our model which are BLUE, ROUGE-L, and METEOR. We applied the same hyperparameters as Ahmad et al. (2020), they are listed in Appendix A.

## 5.2 RESULTS AND ANALYSIS

**Results on Java and Python datasets.** Table 1 shows comparisons of our model GN-Transformer with previous works in code summarization. Our method outperformed all previous works in all metrics. The most suitable comparison of our approach from all previous works is that of a vanilla Transformer rather than Transformer (full). The only difference between our work and vanilla Transformer by Ahmad et al. (2020) is the GN-Transformer Encoder block and the SCG structure, which is the scope of this paper. In contrast Transformer (full) makes use of positional embedding and copy mechanism introduced by See et al. (2017); Shaw et al. (2018). Such improvements apply to all sequence models irrespective of the domain but they do not directly apply to a graph structure. Analogous techniques introduced to Graph Networks should correspond to improvements in our model. However, we consider this to be outside the scope of this paper. Our experiments on positioning encoding and copy mechanism are presented in Appendix C.

We applied the same hyperparameters and network configurations with the vanilla Transformer as proposed in Ahmad et al. (2020). The result shows an improvement of 1.49, 2.09 BLEU, 0.51, 1.10 METEOR and 1.99, 2.34 ROUGE-L in Java and Python dataset respectively. Comparisons with more baselines are shown in Table 2 in Appendix C.

**Additional experiments on Java dataset** We conducted additional experiments in the following three directions on the Java dataset. Results are shown in Table 3. Appendix D provided more details about additional experiment settings.

(A) Hyper parameters. Experiments on different combinations of hyper parameters including the number of layers, embedding size, width of FFN, and configurations of attention heads.

(B) Model structure. We firstly tested a 2-hop GN-Transformer block by using two MHA sublayers in each block thus aggregating two hops of information. Result shows that it harms performance. Collecting two hops of information without a FFN in between is less expressive. We then did an ablation study to show the advantage of the GN-Transformer blocks. We replaced a GN-Transformer block with Graph Attention Networks (GAT) layer with the same hyperparameters as that of a base model. GAT did not perform optimally in our problem when modeled text sequence as a graph. Result shows that the GN-Transformer block largely outperformed GAT even when we compared configurations with similar parameter numbers as in (A). Then we added FFN to GAT layers for an ablation study on MHA. This was to see the impact of MHA. Results show that MHA brings a sig-

---

[4] Hu et al. (2018a)

[5] Wei et al. (2019a)

[6] Ahmad et al. (2020)

nificant improvement. We thus conclude that both the FFN and MHA used by our GN-Transformer blocks are necessary and greatly improve performance.

(C) Variants of graph structures. We tested two variants of graph structures discussed in Section 3.2. Both variants underperform with Variant 1 being much closer to a SCG when compared to Variant 2. Variant 1 underperforms because it introduces redundant edges and leads to redundant interactions between the AST and tokens. It degrades performance little since the structural information and isolation among tokens introduced by the AST were preserved. For Variant 2, the AST code sequence tokens are fully connected. Isolation among tokens is completely lost which leads to the loss of structural information.

The above experiments explain why Ahmad et al. (2020) did not improve performance when using SBT (Hu et al., 2018a). The AST nodes are fully connected. The structural information from the AST graph is not preserved, similar to Variant 2. Moreover, SBT introduced redundant interactions between AST nodes and token nodes similar to Variant 1. However, Variant 2 still achieved 0.43 BLEU, 0.47 METEOR, 0.78 ROUGE-L improvements compared to a vanilla Transformer which shows the usefulness of AST. From our ablation study on the graph structure variants, we conclude that the cause of degradation is the redundant edges and the loss of structural information.

## 6 RELATED WORKS

Integrating valuable information from graph representations like AST, CFG, PDG has long been the focus of deep learning in code understanding. Alon et al. (2019b) retrieves AST embeddings with random walks then concatenates them with token embeddings and finally aggregates them by an attention mechanism as a context vector. Hu et al. (2018a) proposed Structure-Based Traversal (SBT) that flattens the AST into a sequence. Huo et al. (2020) applied DeepWalk with CNN and LSTM to learn a CFG representation which then concatenated with the source code representation. LeClair et al. (2020) used GCN to learn AST embeddings then concatenated them with source code embeddings. Wang et al. (2020) augmented AST features by adding edge type information on AST that representing the control and data flow, then applied gated GNN to learn this augmented-AST embedding. Most existing methods fuse AST and code information by late fusion (Baltrušaitis et al., 2019) that concatenates embeddings of the two different modalities with few interactions of cross-modal information. Veličković (2019) investigated the possibility of early fusion. They proposed cross-connections between models to enable the sharing of cross-modal information. TreeLSTM (Tai et al., 2015) fused tree information into sequence by a tree-structured LSTM. Another category of methods modeled information sources as heterogeneous graphs that are then processed by GNNs. Yao et al. (2019) constructed heterogeneous graphs of documents and text. Ren & Zhang (2020) modeled text as a heterogeneous graph of topics and entities.

Research on general deep learning frameworks proposed a method to train on a uniform representation and model framework (Battaglia et al., 2018; Gilmer et al., 2017; Wang et al., 2018). Graph Networks unified deep learning models on powerful graph representation which could represent arbitrary relational inductive biases between entities (Battaglia et al., 2018). On the other hand, You et al. (2020) reveals the relation between the graph structure and neural network structure which provided a more comprehensive view on the impact of graph structure in deep learning models.

## 7 CONCLUSION

In this paper, we analyzed the fusion of a sequence and a graph from a novel perspective of graph networks. We proposed our GN-Transformer with Syntax-Code Graph. Our method achieved state of the art in two code summarization datasets. We also did experiments on hyperparameters, model structure, and graph structures. Future works include finding the optimal structure of SCG and fusing supplementary information like CFG. Due to the similarity with Transformer, ideas like masked pretraining, positional encoding, copy mechanisms for Transformers also worth interpreting and implementing in the context of Graph Networks. A decoder designed for graph representation may also improve performance, our method discarded AST node embedding while they naturally contained additional structural information which should be useful to the decoder. The simplicity of our method could expand in other domains that there is a duality in sequence and graph representations.

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

## A  SUMMARY OF HYPER-PARAMETERS

| Hyper-parameters | Value |
|---|---|
| Num of layers | 6 |
| Attention heads | 8 |
| $d_k$ | 64 |
| $d_v$ | 64 |
| $d_{model}$ | 512 |
| Hidden units in FFN | 2048 |
| Dropout rate | 0.2 |
| Optimizer | Adam |
| Initial learning rate | 0.0001 |
| Decay rate | 0.99 |
| Max epoch num | 200 |
| Early stop epochs | 20 |
| Training Batch set | 30 |
| Testing Beam size | 4 |
| Max src. vocab size | 50000 |
| Max tgt. vocab size | 30000 |
| Max Java training code len. | 150 |
| Max Python training code len. | 400 |
| Max Java training summary len. | 50 |
| Max Python training summary len. | 50 |

We used the same hyper-parameters as (Ahmad et al., 2020). For the data preprocessing, the code and summary data will be truncated if they exceed the maximum length. We will first truncate the tokens that exceed the max sequence length. For all AST nodes, if the scope contains a truncated token, this AST node will be truncated. We removed all nodes that have a scope which contains truncated tokens. We set a vocabulary size limit, we store only the highest frequent words. Words that are not in the vocabulary will be recognized as $Unknown\_word$. Our methodology is consistent with Ahmad et al. (2020) with the exception of truncating AST since they do not use that information.

## B  DETAILS OF DATASET AND PREPROCESSING

| Dataset | Java | Python |
|---|---|---|
| Examples - Train | 69593 | 64939 |
| Examples - Validation | 8694 | 21605 |
| Examples - Test | 8689 | 21670 |
| Unique Function Tokens | 66569 | 104839 |
| Unique Summary Tokens | 46859 | 64898 |
| Avg. Function Length | 120.29 | 132.64 |
| Avg. Summary Length | 17.73 | 9.56 |
| Avg. AST Nodes | 50.30 | 70.10 |
| Avg. AST Edges | 45.89 | 68.16 |

The statistics of our preprocessed datasets are shown above. Despite the difference in implementation, we kept our methodology consistent with Ahmad et al. (2020). We used the same $CamelCase$ and $snake\_case$ tokenizer from Ahmad et al. (2020) to preprocess source code data in both Java and Python datasets. SCG for subtokens is a little bit different, see Figure 6. We also replaced the strings and numbers in source code by '$\langle STR \rangle$' and '$\langle NUM \rangle$'. For the code summary, we used raw corpus for the Java dataset, and used the same method with Wan et al. (2018) to process code summaries in the Python dataset. We used the train/valid/test split from the original corpus for the

Java dataset and we split the Python dataset by 6:2:2. All of the above are consistent with Ahmad et al. (2020).

There are two differences between our preprocessed dataset and Ahmad et al. (2020):

1. Data cleaning: We discarded the samples that cannot be parsed by the compiler. There are 160 out of 87136 and 4894 out of 113108 samples in Java and Python dataset that are discarded respectively. In the Java dataset, there are no samples removed by Ahmad et al. (2020). For the Python dataset, Ahmad et al. (2020) follow the same cleaning process by Wei et al. (2019b). They removed the samples that exceed a length threshold. They remove 20563 out of 113108 samples.

2. Python Dataset: We used the same method for Python as we did for the Java dataset. Ahmad et al. (2020) deleted special characters in the Python source codes while we preserved them. The result is that the average code length of the dataset preprocessed by them is 47.98 while ours is 132.64.

Apart from the above two differences, the preprocessing methodology is consistent with Ahmad et al. (2020) as well as other baselines reported by them.

## C  ADDITIONAL EXPERIMENT RESULTS

Table 2: Other baselines on Java and Python datasets.

| Models | Java | | | Python | | |
| --- | --- | --- | --- | --- | --- | --- |
| | BLEU | METEOR | ROUGE-L | BLEU | METEOR | ROUGE-L |
| CODE-NN [7] | 27.60 | 12.61 | 41.10 | 17.36 | 9.29 | 37.81 |
| Tree2Seq [8] | 37.88 | 22.55 | 51.50 | 20.07 | 8.96 | 35.64 |
| RL+Hybrid2Seq [9] | 38.22 | 22.75 | 51.91 | 19.28 | 9.75 | 39.34 |
| API+CODE [10] | 41.31 | 23.73 | 52.25 | 15.36 | 8.57 | 33.65 |
| GN-Transformer (Ours) | **45.48** | **26.91** | **55.29** | **33.31** | **19.66** | **46.56** |

The comparison between our method and additional baselines are shown in Table 2. Table 3 showed the results of additional experiments. Table 4 showed experimental results on positional encoding and copy mechanism.

For experiments on positional encoding, we applied a learnable absolute positional embedding (**APE**) layer, which is often used on Transformers. We then use the summation of APE and node embeddings fetched from the input embedding layer as the input to the encoder. We tested APE only on token nodes and used padding on AST nodes.

For the experiments on relative positional encoding (**RPE**) on our model, we had to adapt the original definition for sequence models to that of graph representation. When RPE is applied to sequences it requires the sequence nodes to be fully connected, which is not the case for SCG. Instead, we applied a two-layered PGNN (You et al., 2019) that learns relative positional information on a graph to learn a RPE for each node. We used an APE as the input to PGNN. We set an embedding size of 512. We applied the fixed number of 6 anchor sets with 2 copies each instead of an adjusted anchor set number in You et al. (2019). We concatenate RPE and node embeddings fetched from the input embedding layer. We then reduce its dimension to $d_{model}$ by a linear layer and provide that as input to the encoder. Next, we tested applying RPE only on token nodes by replacing the RPE for AST nodes by padding. For the copy mechanism, we tested the same copy mechanism as Ahmad et al. (2020). All hyperparameters are the same and listed in Appendix A.

---

[7](Iyer et al., 2016)

[8](Tai et al., 2015)

[9](Wan et al., 2018)

[10](Hu et al., 2018b)

Table 3: Additional experimental results. Unlisted values are identical to the base model. Parameter number not including the embedding layer.

| | $N$ | $d_{model}$ | $d_{ff}$ | $H$ | $d_k$ | $d_v$ | BLEU | METEOR | ROUGE-L | params ($\times 10^6$) |
|---|---|---|---|---|---|---|---|---|---|---|
| base | 6 | 512 | 2048 | 8 | 64 | 64 | 45.48 | 26.91 | 55.29 | 44.1 |
| | 2 | | | | | | 37.37 | 20.66 | 49.20 | 14.7 |
| | 4 | | | | | | 43.41 | 24.86 | 53.62 | 29.4 |
| | 8 | | | | | | 45.76 | 27.18 | 55.59 | 58.9 |
| | | | | 1 | 512 | 512 | 43.13 | 24.77 | 53.24 | |
| | | | | 4 | 128 | 128 | 44.79 | 26.20 | 54.59 | |
| (A) | | | | 16 | 32 | 32 | 45.52 | 27.00 | 55.39 | |
| | | | | 32 | 16 | 16 | 45.50 | 27.11 | 55.49 | |
| | | 256 | | | | | 40.01 | 22.35 | 51.16 | 22.1 |
| | | 1024 | | | | | 46.19 | 28.02 | 55.86 | 88.2 |
| | | | 1024 | | | | 43.99 | 25.40 | 54.34 | 31.5 |
| | | | 4096 | | | | 45.82 | 27.63 | 55.71 | 69.3 |
| | 2-hop GN-Transformer block | | | | | | 43.90 | 25.24 | 53.49 | 50.4 |
| (B) | GAT | | | | | | 38.51 | 21.14 | 48.39 | 26.8 |
| | GAT with FFN | | | | | | 40.60 | 22.71 | 50.05 | 39.4 |
| (C) | Variant 1 | | | | | | 45.25 | 26.56 | 55.11 | |
| | Variant 2 | | | | | | 44.42 | 26.87 | 54.08 | |

Table 4: Experimental results on positional encoding and copy mechanism. APE/RPE (token) means only applied APE/RPE on token nodes. Parameter number not including the embedding layers.

| | Java | | | Python | | | params (x10^6) |
|---|---|---|---|---|---|---|---|
| | BLEU | METEOR | ROUGE-L | BLEU | METEOR | ROUGE-L | |
| base | **45.48** | 26.91 | 55.29 | **33.33** | **19.67** | 46.57 | 44.1 |
| APE | 44.18 | 25.68 | 54.41 | 29.40 | 17.08 | 43.17 | 44.1 |
| APE (token) | 44.42 | 25.81 | 54.59 | 31.40 | 18.37 | 45.10 | 44.1 |
| RPE | 45.32 | **27.08** | **55.33** | 32.49 | 19.62 | 46.30 | 46.2 |
| RPE (token) | 45.07 | 26.58 | 55.12 | 31.55 | 19.09 | 45.70 | 46.2 |
| Copy mech. | 45.03 | 26.50 | 54.91 | 33.04 | 19.64 | **46.59** | 44.1 |

The results show that APE will harm the performance of both Java and Python dataset. When applying APE only in token nodes, the degradation is minor. We hypothesize that this could be because APE is not useful for AST nodes, APE in token nodes represent their absolute positions in the input sequence. On the contrary, the positions for AST nodes in the input sequence should be the scope instead of a single absolute position. The absolute positional information is not useful in a standard SCG for the AST nodes.

The results for RPE are also not promising in the interpretation we made for graph representations. We chose a fixed number of anchor sets for all graphs. While in You et al. (2019), they dynamically choosing anchor sets number for different sizes of graphs. Our anchor sets number may be inadequate to capture accurate relative positional information in large graphs and it's revealed by the fact that the average node number of the Python dataset is 70.10 compared to 50.30 in the Java dataset. This reflects on the results since we obtain improvement for some metrics in the Java dataset while performance degradation for Python.

Both positional encoding and copy mechanism did not improve the performance across all metrics but were close to the base model. We hypothesize that both mechanisms have shortcomings when applied for graphs and SCG. Our results show we can not apply tricks that are designed for Transformer in our model without further consideration of the problem domain and graph modality. The ideas found in sequence models can still be designed to fit the graph representation but require additional investigation.

# D DETAILS OF ADDITIONAL EXPERIMENTS

Here we showed more details about our additional experiments discussed in Section 5.2.

For additional experiments (B). We used the GAT layer implementation from DGL[11]. The difference between MHA in GAT and GN-Transformer is that GN-Transformer used the multiplication of features of two nodes in an edge to calculate edge feature while GAT used summation in DGL's implementation. Secondly, Transformer applied four linear layers on key, query, value, and output of MHA respectively while GAT only used one on input features.

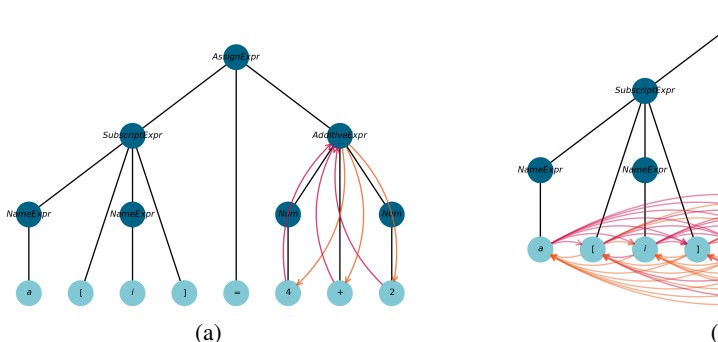

<table>
<tr><td align="center">(a)</td><td align="center">(b)</td></tr>
</table>

Figure 5: (a) Introducing shortcut edges for AST node 'AdditiveExpr' in Variant 1, the three token nodes '4', '+', '2' will connect with it. We marked token nodes by shallow blue, AST nodes by deep blue, edges from token nodes to AST nodes by red, orange for the converse direction. (b) In Variant 2, all token nodes are fully connected with each other.

For additional experiments (C). In Variant 1, we solve the long range dependence problem discussed in Section 3.2 by introducing shortcut edges between each AST node and the token nodes that are related to this AST node. The distance between AST nodes and all nodes within its scope are shortened to 1. It doesn't break any isolation but introduced additional direct interactions. We define the related token nodes of an AST node as all token nodes within the scope of the AST node. Figure 5(a) shows how shortcut edges are added for the 'AdditiveExpr' node, we add bi-directional shortcut edges between the AST node 'AdditiveExpr' and token nodes '4', '+', '2' which are within the scope of this AST node. The edges allow for the AST node to propagate information through the node and edge update rules. This can ameliorate the long range dependence problem in the graph. In the second variant, we make the token nodes fully connected as the Transformer. The nodes are not isolated from each other anymore. This causes all nodes to interact with each other. See Figure 5(b).

# E EXAMPLES OF GENERATED SUMMARIZATION

Here we presented ten examples of the summarization generated by our model and the baselines on the test set. The first five ones are Java examples, the remaining five are Python examples.

```java
public boolean is_selected ( ItemSelectionChoice p_choice ){
    return   sel_array [ p_choice . ordinal () ];
}
```

**Reference:** looks, if the input item type is selected.

**GN-Transformer:** whether a choice is selected, or not

**Transformer:** get if the selection should be a selection

---

[11]https://www.dgl.ai/

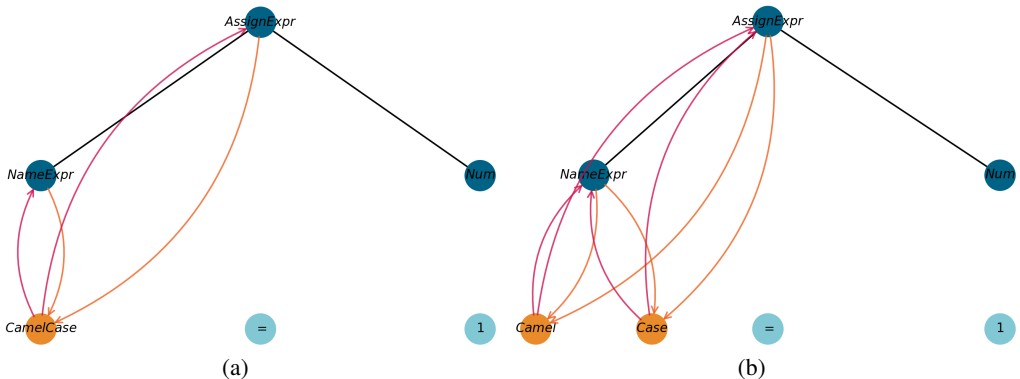

Figure 6: Example of building connections for subtokens. Edges between other tokens and AST are omitted. The figure also shows how shortcut edges are connected to subtoken nodes. (a) Connection for original token node. (b) Connection for subtoken nodes. All subtoken nodes copied the same edges from the original token node.

**Transformer (full):** is a selection find possible

```
private  static  boolean  equalsHandlesNulls(Object a,Object b){
    return  a == b || (a != null && a.equals(b));
}
```

**Reference:** returns true if a and b are equal or are both null .

**GN-Transformer:** returns true if two possibly - null objects are equal .

**Transformer:** tell whether two objects are equal .

**Transformer (full):** checks equality of two objects . returns true if one possibly null .

```
public  static  String  formatListToString ( List <String> list ){
    String  result ="";
    for ( String s :  list )  result +=s + "\t\t";
    result = result . trim () ;
    return  result .toUpperCase();
}
```

**Reference:** formats the list objects from the pokemodel into formatted strings that are easily readable .

**GN-Transformer:** converts a list of strings to upper case strings .

**Transformer:** returns a comma - separated list of strings .

**Transformer (full):** encloses a list of strings from standard separator

```
public  final  boolean  isExceptionHandlerEquivalent (BasicBlock other){
    if ( exceptionHandlers != other . exceptionHandlers )  {
        Enumeration<BasicBlock> e1=getExceptionHandlers();
        Enumeration<BasicBlock> e2=other.getExceptionHandlers();
        while (e1.hasMoreElements()) {
```

```
            if  (!e2.hasMoreElements())  return   false ;
            if  (e1.nextElement()  != e2.nextElement())  return   false ;
        }
        if  (e2.hasMoreElements())  return   false ;
    }
    return   true ;
}
```

**Reference:** compare the in scope exception handlers of two blocks.

**GN-Transformer:** return true if the block is equivalent to the exception handlers, or false otherwise.

**Transformer:** determines whether this exception has an exception?

**Transformer (full):** returns true if this instruction is an exception .

```
protected   static   void  quickSort ( Instances   insts , int [] indices , int   attidx , int   left , int   right ){
    if  ( left  < right )  {
        int  middle= partition ( insts , indices , attidx , left , right );
        quickSort ( insts , indices , attidx , left ,middle);
        quickSort ( insts , indices , attidx ,middle + 1, right );
    }
}
```

**Reference:** sorts the instances according to the given attribute/dimension. the sorting is done on the master index array and not on the actual instances object.

**GN-Transformer:** sorts the specified range of the array using the specified items

**Transformer:** src the ordinal field array into ascending order

**Transformer (full):** sorts the specified range of the array using the given workspace array .

```
def  is_power2 (num):
    return  ( isinstance (num, numbers. Integral )  and  (num > 0) and (not  (num & (num − 1))))
```

**Reference:** test if num is a positive integer power of 2 .

**GN-Transformer:** return true if the power of 2 .

**Transformer:** returns true if and number is a user-defined power .

**Transformer (full):** return whether or not the argument is a power .

```
def  delete (queue, items , backend=' sqlite ' ):
    queue_funcs = salt . loader .queues( __opts__ )
    cmd = '{0}. delete ' . format (backend)
    if  (cmd not in  queue_funcs):
        raise   SaltInvocationError (' Function  "{0}" is  not  available ' . format (cmd))
    ret  = queue_funcs [cmd](items=items, queue=queue)
    return   ret
```

**Reference:** delete an item or items from a queue cli example: .

**GN-Transformer:** delete one or more or all items from a queue cli example: .

**Transformer:** delete an item from a queue cli example: .

**Transformer (full):** delete message(s) from a queue cli example: .

```
def issues_closed_since (period=timedelta(days=365), project='ipython/ipython', pulls=False):
    which = ('pulls' if pulls else 'issues')
    if isinstance (period, timedelta):
        since = round_hour((datetime.utcnow() - period))
    else:
        since = period
    url = ('https://api.github.com/repos/%s/%s?state=closed&sort=updated&since=%s&per_page=%i'
        % (project, which, since.strftime (ISO8601), PER_PAGE))
    allclosed = get_paged_request (url, headers=make_auth_header())
    filtered = [i for i in allclosed if (_parse_datetime (i['closed_at']) > since)]
    if pulls:
        filtered = [i for i in filtered if (_parse_datetime (i['merged_at']) > since)]
        filtered = [i for i in filtered if (i['base']['ref'] == 'master')]
    else:
        filtered = [i for i in filtered if (not is_pull_request (i))]
    return filtered
```

**Reference:** get all issues closed since a particular point in time .

**GN-Transformer:** return a list of closed issues .

**Transformer:** find/download for constructed tol by several /viewfinder convention .

**Transformer (full):** return a list of closed since promotions time .

```
def _mergeOptions(inputOptions, overrideOptions):
    if inputOptions.pickledOptions:
        try:
            inputOptions = base64unpickle(inputOptions.pickledOptions)
        except Exception as ex:
            errMsg = ("provided invalid value '%s' for option '--pickled-options'" %
                inputOptions.pickledOptions)
            errMsg += (("('%s')" % ex) if ex.message else '')
            raise SqlmapSyntaxException(errMsg)
        if inputOptions.configFile:
            configFileParser (inputOptions.configFile)
    if hasattr (inputOptions, 'items'):
        inputOptionsItems = inputOptions.items()
    else:
        inputOptionsItems = inputOptions.__dict__.items()
    for (key, value) in inputOptionsItems:
        if ((key not in conf) or (value not in (None, False)) or overrideOptions):
            conf[key] = value
    for (key, value) in conf.items():
        if (value is not None):
            kb.explicitSettings.add(key)
    for (key, value) in defaults.items():
        if (hasattr (conf, key) and (conf[key] is None)):
            conf[key] = value
    _ = {}
    for (key, value) in os.environ.items():
        if key.upper().startswith (SQLMAP_ENVIRONMENT_PREFIX):
            _[key[len(SQLMAP_ENVIRONMENT_PREFIX):].upper()] = value
    types_ = {}
    for group in optDict.keys():
        types_.update(optDict[group])
    for key in conf:
        if ((key.upper() in _) and (key in types_)):
            value = _[key.upper()]
            if (types_[key] == OPTION_TYPE.BOOLEAN):
```

```
                    try :
                        value  = bool( value )
                    except  ValueError :
                        value  = False
                elif   ( types_[key]  == OPTION_TYPE.INTEGER):
                    try :
                        value  = int ( value )
                    except  ValueError :
                        value  = 0
                elif   ( types_[key]  == OPTION_TYPE.FLOAT):
                    try :
                        value  = float ( value )
                    except  ValueError :
                        value  = 0.0
                conf[key]  = value
        mergedOptions.update(conf)
```

**Reference:** merge command line options with configuration file and default options .

**GN-Transformer:** merges options from a config file .

**Transformer:** merges all of the data used into an option .

**Transformer (full):** loads configuration attributes and add attributes .

```
def  LoadFromString(yaml_doc, product_yaml_key,   required_client_values ,   optional_product_values ):
    if  (_PY_VERSION_MAJOR == 2):
        if  (( _PY_VERSION_MINOR == 7) and (_PY_VERSION_MICRO < 9)):
            _logger .warning(_DEPRECATED_VERSION_TEMPLATE, _PY_VERSION_MAJOR,
                _PY_VERSION_MINOR, _PY_VERSION_MICRO)
        elif  ( _PY_VERSION_MINOR < 7):
            _logger .warning(_DEPRECATED_VERSION_TEMPLATE, _PY_VERSION_MAJOR,
                _PY_VERSION_MINOR, _PY_VERSION_MICRO)
    data  = (yaml. safe_load (yaml_doc) or  {})
    try :
        product_data  = data [product_yaml_key]
    except  KeyError:
        raise  googleads. errors .GoogleAdsValueError(('The "%s" configuration  is  missing' %
            (product_yaml_key,) ))
    if  (not  isinstance ( product_data ,  dict )):
        raise  googleads. errors .GoogleAdsValueError(('The "%s" configuration  is  empty or  invalid '
            % (product_yaml_key,) ))
     IncludeUtilitiesInUserAgent  (data . get (_UTILITY_REGISTER_YAML_KEY, True))
     original_keys  = list ( product_data .keys() )
     client_kwargs  = {}
    try :
        for  key  in   required_client_values  :
            client_kwargs [key]  = product_data [key]
            del  product_data [key]
    except  KeyError:
        raise  googleads. errors .GoogleAdsValueError(('Some of the  required  values  are  missing.
            Required  values  are :  %s, actual  values  are  %s' % ( required_client_values ,
            original_keys )))
    proxy_config_data  = data . get (_PROXY_CONFIG_KEY, {})
    proxy_config  = _ExtractProxyConfig (product_yaml_key,  proxy_config_data )
    client_kwargs ['proxy_config'] = proxy_config
    client_kwargs [' oauth2_client ']  = _ExtractOAuth2Client (product_yaml_key,  product_data ,
        proxy_config )
    client_kwargs [ENABLE_COMPRESSION_KEY] = data.get(ENABLE_COMPRESSION_KEY, False)
    for  value  in   optional_product_values  :
        if  (value  in  product_data ):
            client_kwargs [value]  = product_data [value]
            del  product_data [value]
```

```
if  product_data :
    warnings.warn((’Could not  recognize  the  following  keys:  %s. They were ignored.’  %
        (product_data,)),  stacklevel =3)
return   client_kwargs
```

**Reference:** loads the data necessary for instantiating a client from file storage .

**GN-Transformer:** loads the header data necessary for instantiating .

**Transformer:** loads a data necessary for credit two types .

**Transformer (full):** loads key: data from a loader_context .

## F    ATTENTION VISUALIZATION FOR SYNTAX-CODE GRAPH

The attention visualization for this short program is shown here, we could get more insights about how each nodes paid attention to its neighboring nodes:

```
public   StartListener (Object  resource ) {
    resource=resource ;
}
```

The first figure is attention visualization for standard SCG. We marked the edges from token nodes to AST nodes by red, edges from AST nodes to token nodes by black, edges within AST by orange and red. The remaining figures are attention visualization based on Variant 1. Each figure shows the attention visualization of the nodes in each layer of AST. The edges from token nodes to AST nodes are marked by red, edges from AST nodes to token nodes by black, edges within AST nodes by orange.

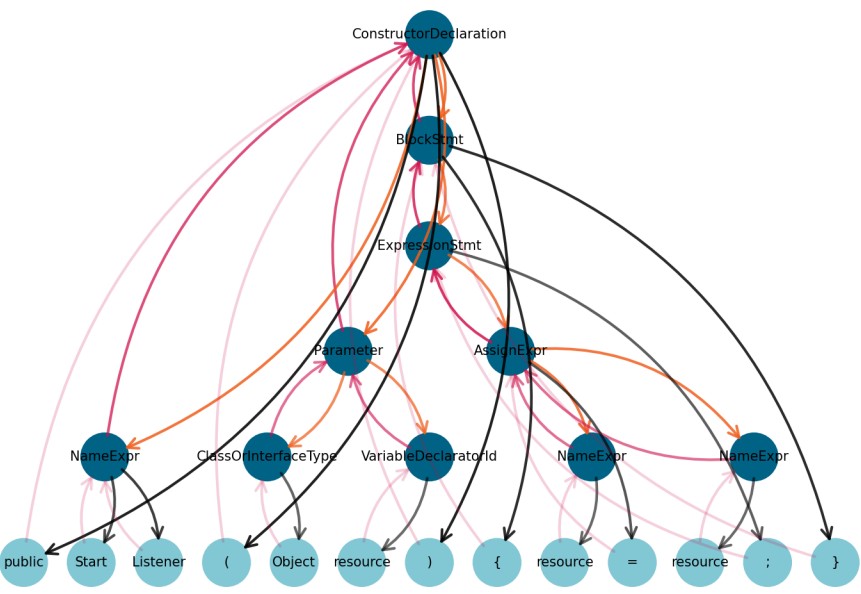

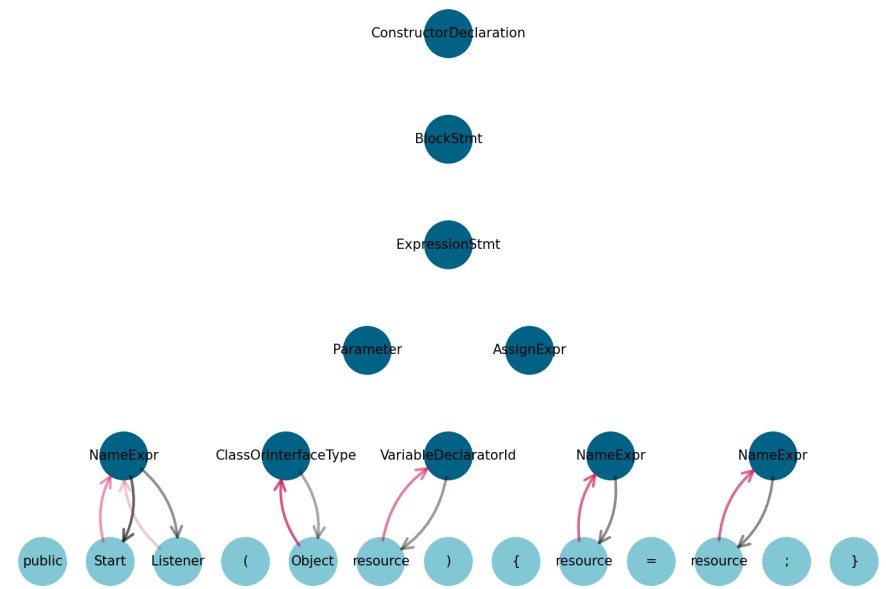

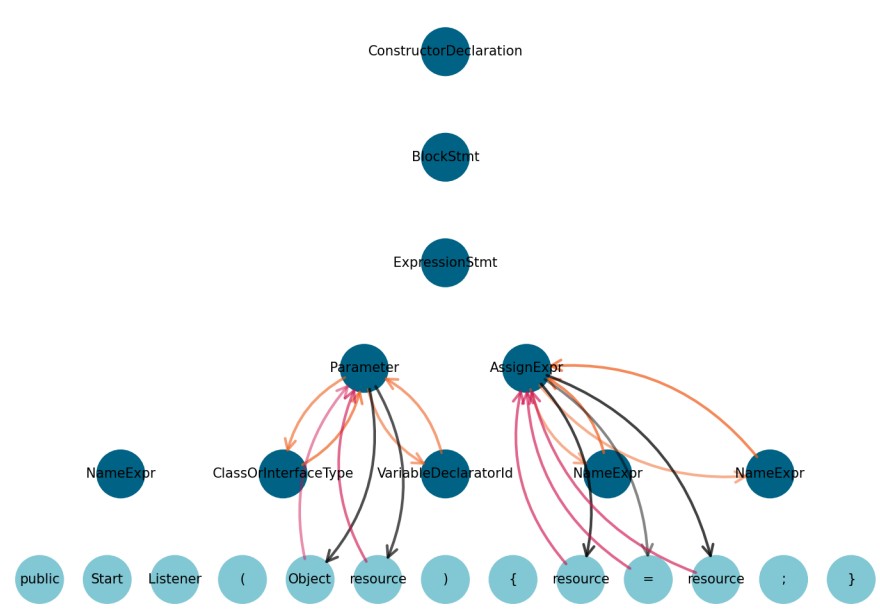

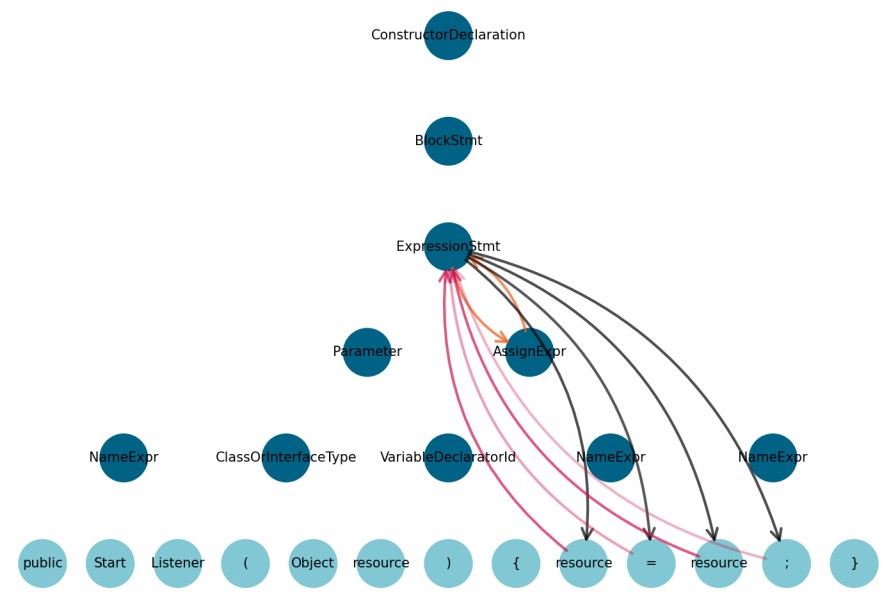

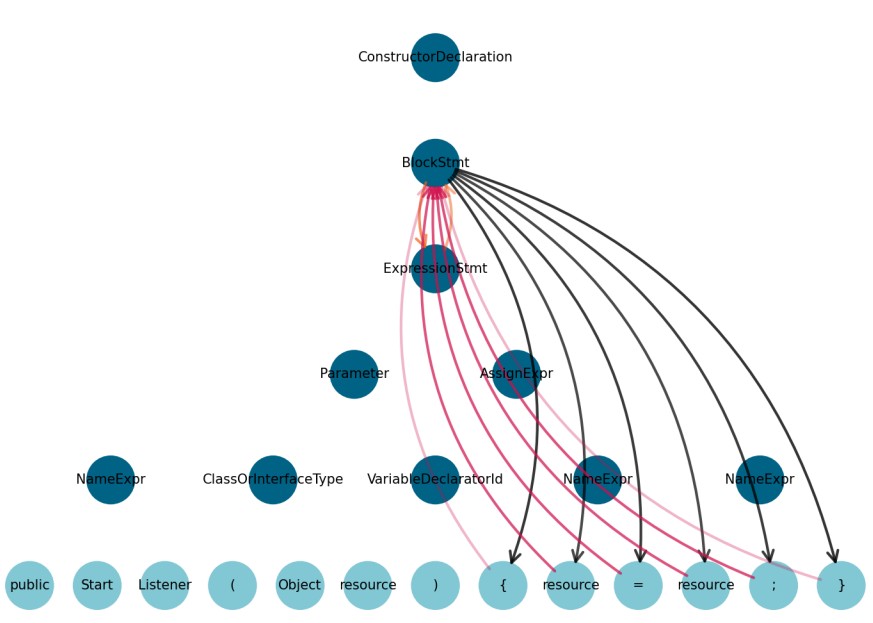

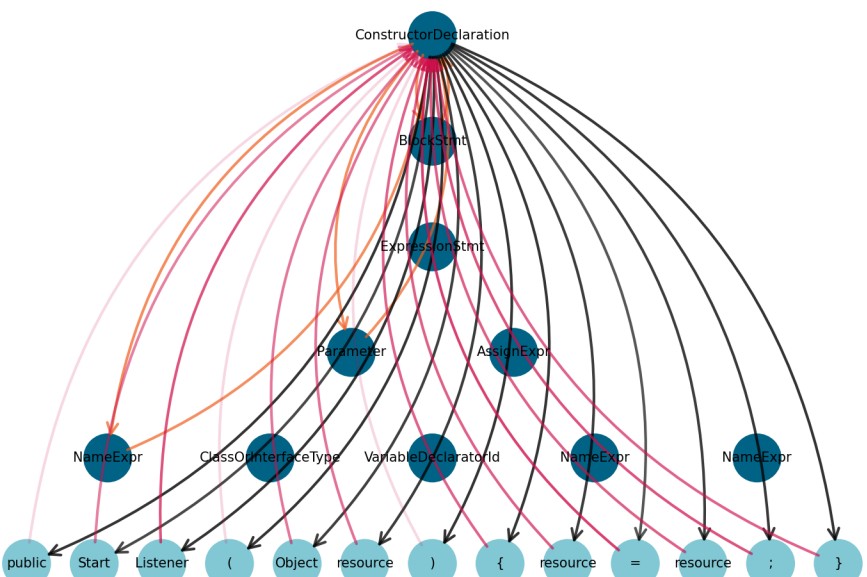

