# OpenReview forum: "GN-Transformer: Fusing AST and Source Code information in Graph Networks"
_ICLR.cc/2021/Conference — Reject_

### Official Review · AnonReviewer3 · 2020-10-26
**Misses significant prior work, and makes some unsupported claims**

**Rating:** 3
**Confidence:** 4

**Review:**

### Summary

This paper focuses on the problem of training a neural model to understand source code. The authors argue that both graph information (such as the parsed abstract syntax tree) and sequence information (such as the raw program tokens) are useful for understanding code, and describe a particular method of adding raw program tokens to a graph called SCS. They also describe a modification of a transformer (called a GN-Transformer) that uses this SCS graph representation, and present results on code summarization tasks.

The proposed encoding does not seem novel, and the authors appear to be unaware of significant prior work in this space. In particular, the SCS graph representation seems like a simpler version of the representation described by Allemanis et al. (2018) [1]. The GN-Transformer model also seems to have only minor differences from previously-used architectures for code understanding. Additionally, the paper has multiple issues with clarity and style that make it hard to understand, and also has some claims that seem unsupported by the evidence.

The experimental results are somewhat interesting, and show that this kind of method can achieve good results on source code summarization tasks (whereas most of the closely-related prior work that I am aware of has instead focused on automatic fixing of bugs). Overall, however, given the lack of novelty of the proposed method and the issues with clarity, I recommend rejection for this paper.


### Detailed comments

In "Learning to Represent Programs with Graphs" [1] (see section 4 heading "Program Graphs"), Allamanis et al. describe a graph representation very similar to the SCS representation described here, which includes both raw tokens and AST nodes, linking the AST nodes to the raw tokens they contain, and connecting each token to the previous and following tokens. The graph representation in [1] also includes a large variety of additional edges with explicit edge labels, based on static analyses of program behaviors. The SCS representation seems to be simpler than this, only including AST and token edges, and not including next/previous relationships for tokens (although the authors do describe a variant where each token is connected to all other tokens) or having any notion of different edge types or directed edges.

On page 3, the authors claim that "structural information contained in the AST is lost" when represented with a structure-based traversal. However, Hu et al. state that the SBT traversal is lossless and that the AST can be recovered exactly from it, so this claim seems to be incorrect.

I don't understand the discussion of "optimal graph structure" in section 3.2. The authors cite work by You et al. (2020), which describes a way to analyze the structure of the connectivity of arbitrary neural networks (including feedforward nets, convolutional nets, etc); they then analyze various properties of these graphs to draw conclusions about the corresponding networks, and find that certain connectivity patterns work better than others. I don't see how this argument translates to graph neural networks, for which the graph is the input to the model. It's not obvious that connectivity of network layers and connectivity of the input graph are the same kind of thing, and also just because connectivity correlates with accuracy in some sense on one dataset that doesn't mean that those connectivity patterns are *necessary* to achieve good accuracy. Furthermore figure 4b in this work suggests that the SCG encoding has a clustering coefficient of exactly 0, which is strange and does not seem to be the same kind of connectivity pattern analyzed by You et al.

The description of the GN-Transformer model is cluttered and hard to follow, but from my understanding it is only a minor variation of existing models. In particular, it appears to be the same as a vanilla transformer, but with masking applied so that each graph node can only attend to neighboring nodes. (Or, equivalently, it is a standard multi-head GAT block except that dot-product attention is used instead of leaky ReLU attention, and a dense feedforward layer is added after the attention mechanism.) There are also similarities to the GREAT model described in "Global relational models of source code" [2], which is also a transformer-based model applied to nodes of an "early-fusion" graph (containing source tokens and AST nodes), but the GREAT model seems more powerful because it is allowed to attend to nodes that are not neighbors (avoiding the long-range dependency problem that the authors of this paper mention in section 3.2).

Also, if my understanding of the GN-Transformer model is correct, I don't see why it is necessary to invoke the formalism of GN blocks (proposed by Battaglia et al). The "edge update function" seems to discard all previous information about each edge, and is just a more complicated way of writing the dot-product attention computation. While it may be true that GN-Transformer fits into the GN block framework, I think using the terminology of GN blocks obscures what the proposed model is doing.

The experimental results seem somewhat promising, and suggest that using graph structure and source tokens together when doing code summarization works better than just operating on sequences or doing a late-fusion approach. However, it seems to me that you could likely obtain even better performance by using a more powerful graph representation and model such as the GREAT model in [2]; it would be important to compare against that approach to determine what, if anything, the SCS representation and GT-Transformer models add.

The claim that the experiments with graph variants "explain" the results of Ahmad et al. (2020) seems too strong. These experiments are conducted with an entirely different architecture and input structure. Maybe these experiments suggest a possible explanation, but there could be many other alternate explanations.

The paper contains some grammatical and notational errors; please carefully reread for these. I have listed some of the things I noticed below, although there may be others.

### Minor errors

Introduction: The sentence "Programming languages are context-free formal language, an unambiguous representation, Abstract Syntax Tree (AST), could be derived from a source code snippet" seems grammatically incorrect.

Figure 2 caption: Some quotation marks face the wrong direction.

Last paragraph of 2.1: some of these sentences are sentence fragments.

Third paragraph of 2.2 "it’s a representation without noise on how the tokens interact": What do you mean by noise here?

Second paragraph of 3.2 "For example, One graph neural networks (GNN) layer": typo in capitalization

First paragraph of 4.2 "This is done by two sub-blocks an edge block": should there be punctuation between "sub-blocks" and "an"?

Second paragraph of 4.2: The notation used here seems inconsistent; $\phi^e$ is defined but never use it, then ${E'_{ij}}^{(\gamma)}$ is described and used but never defined. Are those the same?


### References

[1] Allamanis, Miltiadis, Marc Brockschmidt, and Mahmoud Khademi. "Learning to represent programs with graphs." International Conference on Learning Representations. 2018.

[2]: Hellendoorn, Vincent J., et al. "Global relational models of source code." International Conference on Learning Representations. 2019.

---

### Official Review · AnonReviewer1 · 2020-10-27
**An interesting topic but remains to be improved.**

**Rating:** 5
**Confidence:** 4

**Review:**

This paper discusses the analysis for AST (program code) by graph network-based transformer. This paper is well-motivated but the organization and presentation remain to be improved. Regarding the methodology, it is too simple but still makes sense.

Major Concerns:
1. The presentation should be improved. For example, the motivations are revealed to the reader in Sec. 3.2. I suggest the authors re-organize and rephrase this paper for better presentation. To be honest, this paper is hard to follow, though the idea is good.
2. The methodology though is effective but too simple. It seems this paper only makes an incremental contribution to combine GN and Transformer. I suggest the authors focus on the tasks, more specifically. Because the experimental results only serve as marginal improvements relative to the Transformer, I think I confirmed the oversimplified property of this methodology.

Minor Concerns:
1. Fig.2: Very confusing figure, because you shall mention 50000 in the caption and specify the process in the caption also. Because of weak description, This paper is hard to read.
2. Fig.3: What is the meaning of showing (a)? The entire Fig.3 seems redundant to me.
3. Did you perform the significance test for the experimental results? Because your improvements are marginal.

Discussion:

This paper really proposed an interesting idea. But for me, it seems like an end-to-end framework to transform tree-based to sequential texts. I have a question: Will the description accurate if we don't see the whole picture of the codes? I just want to provide a new perspective for you to look into.

---

### Official Review · AnonReviewer2 · 2020-10-28
**lack of novelty, paper writing needs to be improved, the broad claim of the title cannot be supported by the only task conducted in the experiment**

**Rating:** 5
**Confidence:** 3

**Review:**

##########################################################################

Summary:


The paper provides a interesting direction in representing source code. In particular, it proposes a method called GN-Transformer to fuse representations learned from graph and text modalities under the Graph Networks.
The proposed method achieved state of the art performance in two code summarization datasets.

##########################################################################

Reasons for score:


Overall, I vote for rejection. I like the idea of injecting code structure in the code representation learning process. My major concern is about the novelty of the paper. Even though the figure looks fancy, the methodology itself seems only standard mask attention. A more recent work has explored ways towards the integration of tree into Transformer, which should be considered. https://arxiv.org/pdf/2002.08046.pdf. Moreover, the claim of "Most existing methods are late fusion and underperform when supplementing the source code text with a graph representation." needs evidence to support. The claim of the title is about a generic method for code representation learning, however, the experiments are only conducted on code summarization task, which is not convincing enough to support such claim. Last, paper writing and organization needs to be improved. Many important details like number of a part of baselines are given in appendix. Appendix should not be used in this way.


##########################################################################

Concerns:


1. My major concern is about the novelty of the paper. Even though the figure looks fancy, the methodology itself seems only standard mask attention. A more recent work has explored ways towards the integration of tree into Transformer, which should be considered. https://arxiv.org/pdf/2002.08046.pdf.

2. The claim of "Most existing methods are late fusion and underperform when supplementing the source code text with a graph representation." needs evidence to support.

3. The claim of the title is about a generic method for code representation learning, however, the experiments are only conducted on code summarization task, which is not convincing enough to support such claim.

4. Paper writing and organization needs to be improved. Many important details like number of a part of baselines are given in appendix. Appendix should not be used in this way.

---

### Official Review · AnonReviewer4 · 2020-10-29
**The study offers an interesting direction for solving the problem, however, it falls short in providing enough evidence on the value of the proposed SCG representation and designed architecture.**

**Rating:** 5
**Confidence:** 4

**Review:**

Summary:

 In this work, authors propose a new direction in summarizing code snippets by combining their AST and lexical code features in the form of a graph called SCG (which is shown not to be optimal). Their model, GN-Transformer, further extracts information from SCG to summarize the code snippet. This model is a combination of transformers and graph network and is benchmarked on two datasets of Java and Python source code. The results show that the model marginally outperforms the selected benchmarks.

%%%%%%%%%%%%%%%%%%%%%%%%%%%%%%%%%%%%%%%%%%%%%%%%%%

Strengths of the paper:

The idea of combining syntactic features of the code (AST) with its lexical feature into a single graph structure is interesting.

The related work section is strong and captures the majority of recent work on the topic.

The paper is well-written and easy to follow. It is well-motivated and unfolds the intuition behind the arguments clearly.



%%%%%%%%%%%%%%%%%%%%%%%%%%%%%%%%%%%%%%%%%%%%%%%%%%

Weakness of the paper:

See detailed review below.

%%%%%%%%%%%%%%%%%%%%%%%%%%%%%%%%%%%%%%%%%%%%%%%%%%



Detailed Review:



Major concerns:

The results of the experiments are rather underwhelming and show marginal improvement compared to the selected benchmarks, specifically the one from Ahmad et al. As such, it is not evident how practical the proposed method is in real-world scenarios. It also does not support the claim by the authors that considering SCG design adds more value compared to capturing AST and sequence information separately.

I find the choice of baselines rather limited. The paper has a rich related work study. However, the focus of the paper is put on the comparison with the work of Ahmad et al.. Many similar research mentioned by authors are not included as baselines. For example, it is far more interesting to see how the SCG-based summarization can compete with CFG-based model of Huo et al., than seeing the comparison with baselines in Appendix C. This can help with proving the value of the model under practical setting in which variety of options are available and not only those based on AST and sequence modeling.

The design of SCG (which is not optimal as authors mentioned as well) does not seem to help much with improving the results. Authors discuss possible ways to improve SCG (e.g. hand-engineering or the use of additional rules), but do not implement them in their experiments. Why is that? Their argument about finding the “sweet spot” between C and L does not go into details of how it can be done and where that “spot” lies in.

In section 3.2, it is mentioned the average path length L in Python and Java corpus is above 6. However, we see L for test data is around 4. I think it is more realistic to test on scripts with L smaller and larger than the average L in the training data to see how it affects the results.

In section 3.2 where the two variants are introduces to alleviate the problems of standard SCG, I was wondering why a combination of the two is not tested as well. Why authors did not test that?



 %%%%%%%%%%%%%%%%%%%%%%%%%%%%%



Minor Concerns:

In figure 3c, as far as I understood, the black edges belong to original AST and the red ones are added. However, there are some AST edges that are red in 3c. Is that a mistake or it is intended? If the latter, why?

The term “code features” is broad. A code snippet has lexical, layout, syntactic, and dynamic features. I believe this study uses “code features” to refer to lexical features only, so it is best to specify that.

The paper has to be self-sufficient, so in section 5.1, it is best to have a brief summary of the selected baselines and their shortcomings or at least how they differ from the propose model in the paper.

The fonts in figure 3 are not readable.

Some abbreviations are used before they are defined. For example, FFA and MHA in page 4, and “SBT” in section 5.2.

 Some typos and grammatical errors. Examples:

Figure 1, in the caption: “blocks” -> block

Figure 4b, x-axis: “clustering coefficiency”

Page 5: “could be lead to...”



%%%%%%%%%%%%%%%%%%%%%%%%%%%%%%%%%%%%%%%%%%%%%%%%%%



Questions:



Please answer the questions raised in “detailed review” above.

---

### Decision · Program_Chairs · 2021-01-07
**Final Decision**

**Decision:**

Reject

**Comment:**

While there are some potentially interesting aspects to this work, it doesn’t acknowledge a significant amount of relevant literature, and there are some unsupported claims. All reviewers believe the paper is not ready for acceptance. Reviewers provided some good thorough reviews and suggestions, but the authors did not choose to respond or engage in discussions to improve the paper.